# A Systematic Review on Food Baskets Recommended in the Eastern Mediterranean Region

Mona Pourghaderi [1], Anahita Houshiarrad [2], Morteza Abdollahi [2], Ayoub Al-Jawaldeh [3], Fatemeh Esfarjani [4], Mohammad-Reza Khoshfetrat [4], Ghasem Fadavi [5] and Fatemeh Mohammadi-Nasrabadi [4,*]

1    Health Equity Research Center (HERC), Tehran University of Medical Sciences, Tehran 1416833481, Iran
2    Department of Nutrition Research, National Nutrition and Food Technology Research Institute, Faculty of Nutrition Sciences and Food Technology, Shahid Beheshti University of Medical Sciences, Tehran 1981619573, Iran
3    World Health Organization Regional Office for the Eastern Mediterranean, World Health Organization, Cairo 7608, Egypt
4    Research Department of Food and Nutrition Policy and Planning, National Nutrition and Food Technology Research Institute, Faculty of Nutrition Sciences and Food Technology, Shahid Beheshti University of Medical Sciences, Tehran 1981619573, Iran
5    Food, Halal and Agricultural Products Research Group, Research Center of Food Technology and Agricultural Products, Standard Research Institute, Karaj 3174734563, Iran
*    Correspondence: f.mohammadinasrabadi@sbmu.ac.ir

**Abstract:** To assist in providing a robust regional set of data and international comparisons, a systematic review was conducted to identify and characterize food baskets (FBs) in the Eastern Mediterranean Region (EMR) countries. Electronic databases of peer-reviewed literature, including PubMed, Scopus, ISI/WOS and Google Scholar, and also, online grey literature, were systematically searched from January 2000 to September 2021. The quality of included studies was assessed using the Joanna Briggs Institute's (JBI) Critical Appraisal checklists for analytical cross-sectional studies. A total of 20 studies and reports were identified as eligible for inclusion in this systematic review. Linear & goal programming is used in many studies to estimate the FB groups. According to the recent recommendations based on sustainability, less consumption of red meat is proposed, and the poultry group, along with eggs, plays an important role in supplying animal protein in EMR FBs. More than 30 g of legumes has been suggested based on the dietary habits of this area, whereas consumption of more than 30–40 g of oils and fats will not be appropriate for the region. The research results are not comparable due to differences in the tools, protocols, and methods; hence, there is a need for a standardized regional approach.

**Keywords:** survival/minimum cost/expenditure food basket; healthy/sustainable food basket; systematic review; Eastern Mediterranean Region





## 1. Introduction

A healthy diet is an integral part of the concept of health [1,2]. Dietary risks are among the major causes of death and their effects on diseases and disability are the second leading cause of Disability-Adjusted Life Years (DALYs) worldwide [3]. According to theGlobal Burden of Disease Study, 11 million deaths and 255 million DALYs were attributed to dietary risk factors in 2017 [4]. In addition to conflicts, displaced populations and political instability, people in the Eastern Mediterranean Region (EMR) also suffer from a severe burden of malnutrition. Some countries continue to experience high levels of food insecurity, malnutrition, and micronutrient deficiencies. At the same time, half of the adult population is overweight or obese, and unhealthy diets are the major risk factor for non-communicable diseases, which account for two-thirds of deaths in this region [5].

The focus of global policy on promoting healthier food choices has increased the need for data on comparative components and affordability of healthy foods [6]. Healthy food

baskets (HFBs) are constructed based on the cost of basic healthy eating that represents current nutrition recommendations and average food purchasing patterns to monitor both the affordability and accessibility of foods. They can address and analyze diet-related health inequalities [7–9], and a commitment to using their results could promote a healthy diet at minimal cost for vulnerable groups [10,11]. On the other hand, according to the latest data on environmental degradation, sustainable diets can be achieved through HFB designing, too [12].

A variety of food baskets (FBs) have been developed for a variety of purposes at state, regional, and community levels, including serving as the basis for the maximum nutrition assistant program benefit allotments [13], examining the cost difference of healthy and unhealthy foods [14,15], comparing the price of healthy foods in remote or rural versus metropolitan locations [16,17], mapping the availability of healthy foods in different locations [18–20], calculating the minimum cost of an adequate diet for social policy planning [21], developing educational material on low-cost healthy eating [22], calculating the environmental costs associated with different food patterns [23], examining trends on food costs over time [24,25], and monitoring the changing affordability of a healthy diet compared to income and welfare support [26–28]. Given these different objectives, there have also been a variety of methods employed to define HFBs. Some have developed mathematical optimization models to define the FBs that meet nutrition recommendations for minimum cost [29] while others have restricted the baskets to a few key food groups such as fruits and vegetables, or basic food staples [30,31].

Nutritional adequacy, health promotion, non-communicable disease prevention, sustainability, and cultural acceptance are all limitations that should be considered in the design of FBs [32]. Healthy eating and lifestyle recommendations will not be practical or acceptable unless they address the human need for social activism, too [33]. A review of the determinants of household FB composition revealed three categories of factors affecting the contribution of different food groups in the household FB: demographic, socioeconomic, and environmental. Accordingly, we can say that these factors determine the healthiness ofahousehold diet [34].

To assist in providing a robust regional set of data and conductinginternational comparisons, we conducted a systematic review to determine similarities and differences, as well asthemethods used in designing FBs in EMR countries.

## 2. Materials and Methods

### 2.1. Identification of Relevant Studies

Electronic databases of peer-reviewed literature, including PubMed, Scopus, ISI/WOS, and Google Scholar, as well as online grey literature, were systematically searched from January 2000 to September 2021 using the PRISMA approach (Appendices A and B) [35]. The same search strategy was applied in all electronic databases. Key terms were categorized into three groups and used in combination with each other as follows:

(healthy OR standard* OR affordable OR minimized OR adequate OR "low cost" OR optim* OR sustainable OR reference OR survival OR nutritious OR thrifty OR basic OR balance*) AND ("food basket" OR diet OR "food plan" OR "food aid") AND (AfghanistanOR-Bahrain OR DjiboutiOREgyptORIranORIraqORJordanORKuwaitORLebanonORLibyaORM oroccoORPalestineOROmanORPakistanORQatarOR"SaudiArabia"ORSomaliaOR SudanO RSyriaORTunisiaOR"United Arab Emirates"ORYemen OR "Middle East").

### 2.2. Inclusion and Exclusion Criteria

Articles, protocols, and reports relating to healthy, optimum, sustainable, survival, and minimum expenditure FBs with different costs in EMR were considered. There was no limitation for the target group selection in terms of household size, FB composition, and also, documents in other languages than English. Due to the predominance of the Arabic language in the region, in addition to Persian and English languages, Arabic sites, sources, articles, and documents were also searched and translated. We did not use any automation

tools for the exclusion of records; rather, all of this process was done by the researchers. Search results for the same journal article or web links to the same report were excluded as duplicates. However, discrete journal articles and reports related to the same collected data set were included. After omitting the duplicates, the titles and abstracts of all identified texts were read to exclude those that were irrelevant. Searching and choosing the final eligible studies and reports were conducted independently by two reviewers (F.M.N. and M.P.). Following the identification of pertinent results, reference lists were also reviewed and hand searching identified other known documents. This process added four new records. Missing data required for review were requested by emailing the correspondent authors a maximum of two times with at leastaone-week interval.

*2.3. Data Extraction, Synthesis, and Quality Assessment*

For selected articles and reports, all the data relating to year, country, FB constructing method, source of data, target group, reported results, and components of FBs (food groups in gram), including bread, macaroni or pasta, rice, potato (or starchy vegetable), vegetable, fruit, milk and dairy products, red meat, poultry, fish, egg, legume (or pulse), nut, fat and oil, sugar and sweet were extracted and summarized in data extraction table A descriptive analysis of the findings was performed.

Quality assessment of the included studies was done by using the Joanna Briggs Institute (JBI) Critical Appraisal checklists for analytical cross-sectional studies. Each study was rated high (H), medium (M), or low (L) according to the number of Yes options selected from the checklists. The score ranges of 0–3, 4–6, and above 6 were considered as low, medium, and high-quality studies, respectively [36].

## 3. Results

*3.1. Study Selection Process*

A total of 6457 studies were identified by searching the initial databases and 21 additional records were identified through searching other sources, including websites and organizations, as well as citation searching. After the removal of 5889 duplicates, 568 studies remained. A total of 552 out of the remaining articles did not meet the selection criteria, so they were excluded after screening the titles and abstracts because they were irrelevant. Out of the remaining 28 studies and reports retrieved, eight of them were excluded after reviewing their fulltext because their results did not comply with the objectives of the current study. Finally, 20 studies and reports were eligible for inclusion in this systematic review (Figure 1). Moreover, three FBs recommended by international organizations were reviewed as a basis for comparison.

*3.2. Study Characteristics*

Table 1 shows the components of FBs designed with different costs in EMR, which were included in the review. Most studies (*n* = 7) in the field of optimum FB have been registered from Iran [37–45], and then Lebanon [46–48] and Pakistan [49–51] with three, and Yemen with two studies [52,53].Other countries in the region with one study were as follows: Syria [54], Iraq [55], and Jordan [56]. Except for Iran and Pakistan/Afghanistan, other food baskets developed in the countries of this region are Survival Minimum Expenditure Baskets (SMEB).

Three other included documents in Table 2 are the Guidelines of the UN Refugee Agency and World Food Program (WFP) for the Survival Minimum Expenditure Basket [57,58], and the Global Healthy Reference Diet, which have been proposed to be sustainable [59–61].

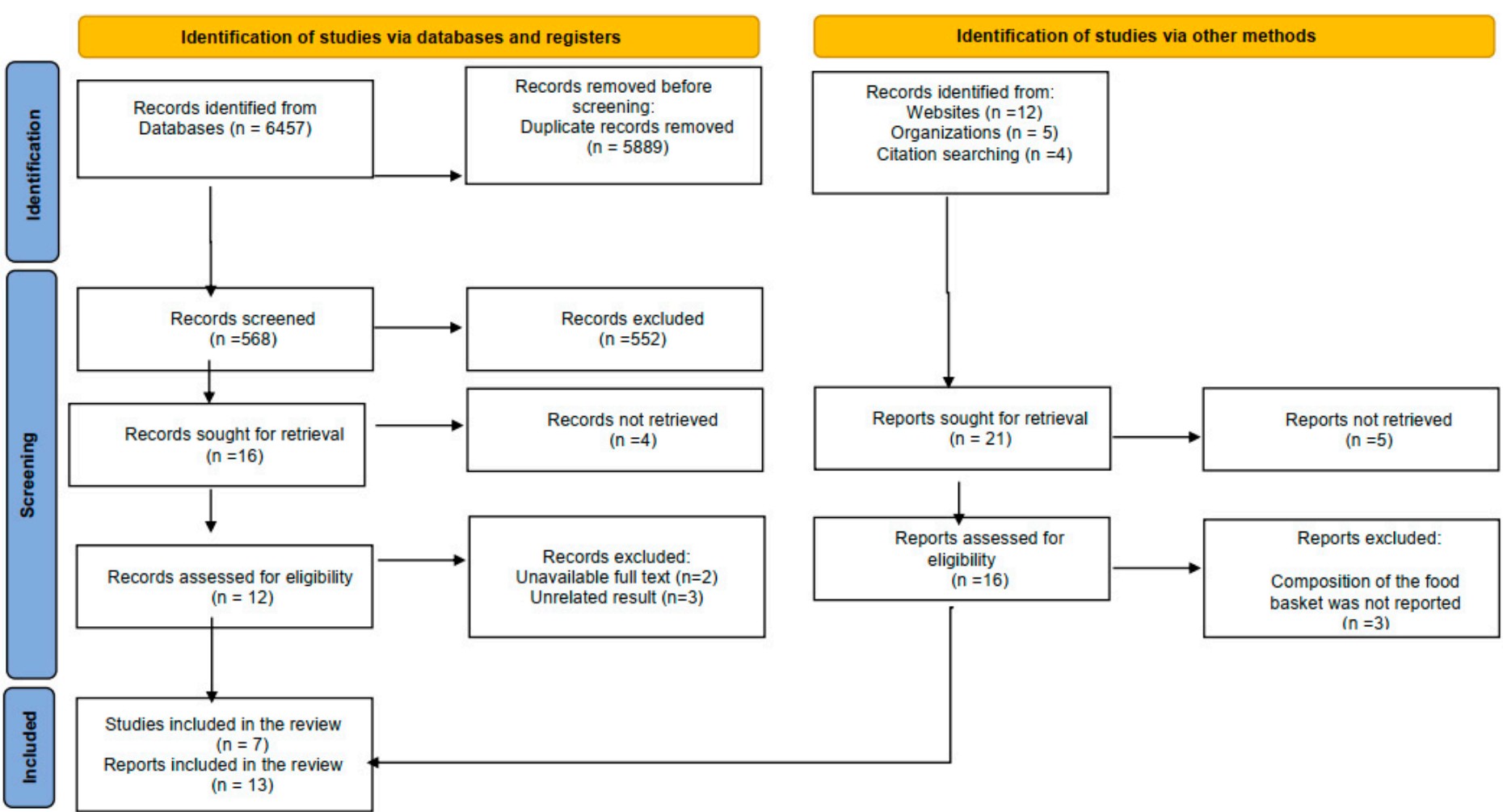

**Figure 1.** PRISMA 2020 flow diagram [35].

**Table 1.** Characteristics of healthy, optimum, sustainable, survival, and minimum expenditure food baskets with different costs in the Eastern Mediterranean Region.

| No. | Author, Date | Constructing Method | Source of Data/Population | Results Based on Energy Provided | | Components of Food Basket (Food Groups in g) | | | | | | | | | | | | | | | | Quality of the Study |
|---|---|---|---|---|---|---|---|---|---|---|---|---|---|---|---|---|---|---|---|---|---|---|
| | | | | | | Bread | Macaroni/Pasta | Rice | Potato (Starchy Veg) | Vegetable | Fruit | Milk &Dairy Products | Red Meat | Poultry | Fish | Egg | Legume (Pulse) | Nut | Fat & Oil | Sugar &Sweet | |
| | | | | | | | | | | Iran | | | | | | | | | | | |
| 1 | Ghasemi et al., 1996 | Min basket providing energy, Max basket providing all nutrients | Food consumption survey, 1991–5 & Income and Expenditure Survey data from SCI [a] 1995 | Per capita daily g in households | Min 2465 Kcal | 350 | 10 | 107 | - | 266 | 169 | 145 | 81 | | | 20 | 18 | - | 34 | 50 | High |
| | | | | | Max 2548 Kcal | 275 | 30 | 120 | - | 350 | 250 | 240 | 110 | | | 15 | 30 | - | 30 | 50 | |
| 2 | Kiani K, et al., 2004 | Linear programming | Food consumption survey, 2001 (Iranian households | Province/2728 Kcal | | 320 | 20 | 100 | 70 | 280 | 260 | 225–240 | 48 | 50 | | 24 | 26 | - | 35–40 | 45–55 | High |
| 3 | Pourkazemi M & Souzandeh M, 2009 | Goal programming | Income and Expenditure Survey data from SCI [a] 2011, Iranians households | 1–3 years 4–6 years 7–9 years 10–14 years 15–18 years 19–50 years >51 years/Sex | Rural: 8399 Kcal, | 196.1 | 87.6 | 91 | 36.6 | 142 | 441.3 | 537 | 10 | 32 | 62.1 | 10.7 | 31.3 | 20 | 38.7 | 11.1 | Medium |
| | | | | | Urban: 9686 Kcal | 148.9 | 146 | 69.3 | 85.5 | 99.3 | 478.7 | 554.6 | 11.6 | 24.6 | 41.5 | 12.9 | 56.9 | 20 | 31.8 | 30.1 | |
| 4 | Salehi F, et al., 2013 | Mean requirement of energy, protein, and key nutrients (Iron, Vitamin A, Riboflavin & Calcium) | Food balance sheet adapted by food consumption coefficients 2011 (7158 Iranian households) | Sex 2–3 years 4–5 years 6–11 years 12–17 years 18–29 years 30–60 years >60 years/ 2573 Kcal | | 310 | 20 | 95 | 70 | 300 | 280 | 250 | 38 | 64 | | 35 | 26 | - | 35 | 40 | High |
| 5 | Nasari A, et al., 2017 | Weighted goal programming | Income and Expenditure Survey data from SCI/Rural Iranians households | Income deciles/2500 Kcal | | 454.1 | | | | 149 | 87.5 | 288.5 | 143.9 | | | 29.4 | 124.8 | - | 75.6 | 87.9 | Medium |
| 6 | Eini-Zinab H, 2021 | Water & carbon food print; linear & goal programming | Income and Expenditure Survey data 1991–2011 from SCI, Iranians households | Adult male/2800 Kcal | | 267.0 | 153.3 | | 77.1 | 207.7 | 256.6 | 231.6 | 7.8 | 81.2 | 4.1 | 10.4 | 34.0 | 5.3 | 44.5 | 72.2 | Medium |

**Table 1.** *Cont.*

| No. | Author, Date | Constructing Method | Source of Data/Population | Results Based on Energy Provided | Bread | Macaroni/Pasta | Rice | Potato (Starchy Veg) | Vegetable | Fruit | Milk &Dairy Products | Red Meat | Poultry | Fish | Egg | Legume (Pulse) | Nut | Fat & Oil | Sugar &Sweet | Quality of the Study |
|---|---|---|---|---|---|---|---|---|---|---|---|---|---|---|---|---|---|---|---|---|
| | | | | | colspan Components of Food Basket (Food Groups in g) | | | | | | | | | | | | | | | |
| 7 | Soltani A, et al., 2020 | SSM-iCrop2 model Plant production with the Water and Production modules | Demand for products to feed the Country as a function of population, diet, food loss and waste | Population diet/2573 Kcal | 364 | | 63 | 109 | 228 | 212 | 190 | 19.1 | 49 | 18 | 25 | 30 | - | 46 | 66 | High |
| | | | | | | | | | | Lebanon | | | | | | | | | | |
| 8 | WFP & UNHCR & UNICEF, 2014 | Cash Working Group discussed and endorsed MEB [f] after consolidation and analyzing | Secondary data on expenditures collected by 17 agencies | WFP ration to meet nutrient needs + 2100 Kcal/month | 70 (+130 Bulgur Wheat) | 50 | 100 | - | 95 | - | 20 | 38 | | | 20 | 60 | - | 33 | 50 | Medium |
| 9 | WFP & UNHCR & UNICEF, 2014 | Cash Working Group discussed and endorsed SMEB [g] after consolidation and analyzing | Secondary data on expenditures collected by 17 agencies | WFP vouchers. Quantities to cover 2100 Kcal/day | 130 (Bulgur Wheat) | 50 | 200 | - | - | - | - | 38 | | | - | 50 | - | 33 | 50 | Medium |
| 10 | El Koury and Hajal. 2016 | FGDs [b] with the refugees who are classified as vulnerable for the quantitative section, item ratings, and item removal | WFP ration to meet nutrient needs | Minimum Food Expenditure Basket per HH [c] 2100 Kcal/month (MEB [d]) | 200 | 50 | 100 | - | 95 | - | 20 | 38 | | | 30 | 60 | - | 33 | 50 | Medium |

**Table 1.** *Cont.*

| No. | Author, Date | Constructing Method | Source of Data/Population | Results Based on Energy Provided | Bread | Macaroni/Pasta | Rice | Potato (Starchy Veg) | Vegetable | Fruit | Milk &Dairy Products | Red Meat | Poultry | Fish | Egg | Legume (Pulse) | Nut | Fat & Oil | Sugar &Sweet | Quality of the Study |
|---|---|---|---|---|---|---|---|---|---|---|---|---|---|---|---|---|---|---|---|---|
| | | | | **Components of Food Basket (Food Groups in g)** | | | | | | | | | | | | | | | | |
| 11 | El Koury and Hajal. 2016 | FGDs with the refugees who are classified as vulnerable for the quantitative section, item ratings, and item removal, | Based on WFP vouchers | The Survival Minimum Expenditure Basket to cover 2100 Kcal/day (SMEB [e]) | 130 | 50 | 200 | - | - | - | - | 38 | | | - | 50 | - | 33 | 50 | Medium |
| 12 | UNHCR, WFP, Save The Children, Relief International, UNICEF, and LOUISE, 2020 | Based on the Survival and Minimum Food Expenditure Basket defined by WFP to meet the minimum nutritional and caloric requirements | Refugee populations in Lebanon | Required NFIs per households of five persons to cover 2100 Kcal/day | 220 +60 (Brown Bulgur) | | 65 | 90 | 20 (Tomato paste) + 10 (Canned Green Pea) + 100 (Cabbage) + 20 (Carrot) | 60 | 10 (Powder Milk) + 10 (Canned Cheese) | 10 | 10 | 10 | 10 | 30 (Lentie) + 10 (White Bean) + 20 (Chickpea) | - | 17 | 20 | High |
| | | | | | | | | | **Pakistan/Afghanistan** | | | | | | | | | | | |
| 13 | Rubin V, 2011 | Focus group discussion was done to identify 'normal consumption patterns and identify key dietary boundaries (LOCAN Diet) [g], Pakistan | Market surveys to identify the lowest cost diet that meets the needs for energy and micronutrients CMWG [h] by | Family includes 2 adults (1 man and 1 woman), and 5 children (Daily Quantity (g)) | 47.3 | - | 184.8 | - | 302.7 | - | 269 | - | 15.7 | - | - | 21.1 | - | 36.6 | 18 | Medium |

**Table 1.** *Cont.*

| No. | Author, Date | Constructing Method | Source of Data/Population | Results Based on Energy Provided | | Bread | Macaroni/Pasta | Rice | Potato (Starchy Veg) | Vegetable | Fruit | Milk &Dairy Products | Red Meat | Poultry | Fish | Egg | Legume (Pulse) | Nut | Fat & Oil | Sugar &Sweet | Quality of the Study |
|---|---|---|---|---|---|---|---|---|---|---|---|---|---|---|---|---|---|---|---|---|---|
| 14 | Ministry of Planning, Development and Reform Planning Commission, 2016 | Estimating the cost of the nutritious diet (CoD) and a staple adjusted nutritious diet by the COD software, Pakistan | The Household Integrated Economic Survey HIES) 2013–2014 | The edible weight and cost of the selected food for family of 6 (the whole year)/average energy need of 2350 Kcal | | 359.8 | - | - | - | 301.2 | 6.2 | 297.4 | - | 35.3 | | | 137.6 | - | 2.3 | 5.9 | Medium |
| 15 | Dizon et al., 2019 | CoRD [i] of achieving the recommended diet based on FBDGs in Afghanistan and Pakistan | The price of each food item and information on FBDGs [j] | An average adult man/99% 2725 Kcal | Min | 280 | | | | 300 | 107 | 200 | 50–90 | | | | 70 | - | 30 | - | High |
| | | | | | Max | 533 | | | | 433 | 213 | 300 | 120–200 | | | | 107 | | 60 | | |
| | | | | | | | | | | Yemen | | | | | | | | | | | |
| 16 | CMWG [h], 2017 | Multi-sectoral market assessment, which covered 97 districts in 12 Governorates | What an average family of seven in Yemen would need, as a minimum, to survive for one month | SMEB (grams/per person/per day) for 1663 Kcal energy need (80% of the monthly household food needs) | | 357 | - | - | - | - | - | - | - | - | - | - | 48 | - | 38 | 12 | Medium |
| 17 | Food Security and Agriculture Cluster (FSAC), 2019 | A series of technical working group meetings | The food commodities market price data | SMEB) for 1676 Kcal energy need (80% of the monthly household food needs) | | 312 | - | - | - | - | - | - | - | - | - | - | 45 | - | 38 | 14 | Medium |
| | | | | | | | | | | Syria | | | | | | | | | | | |

**Table 1.** *Cont.*

| No. | Author, Date | Constructing Method | Source of Data/Population | Results Based on Energy Provided | Components of Food Basket (Food Groups in g) | | | | | | | | | | | | | | | Quality of the Study |
|---|---|---|---|---|---|---|---|---|---|---|---|---|---|---|---|---|---|---|---|---|
| | | | | | Bread | Macaroni/Pasta | Rice | Potato (Starchy Veg) | Vegetable | Fruit | Milk &Dairy Products | Red Meat | Poultry | Fish | Egg | Legume (Pulse) | Nut | Fat & Oil | Sugar &Sweet | |
| 18 | Cash Based Responses Technical Working Group Syria, 2014 | Basic survival commodities as a criterion & standardized process for determining the value of the SMEB | Food commodities in northern Syria | Recommended daily energy requirements of 2100 Kcals per person per day | 200 | 80 | 100 | - | 30 | - | - | 30 | | | 30 | 100 | - | 40 | 25 | Medium |
| | | | | | | | | | Iraq | | | | | | | | | | | |
| 19 | Cash Working Group, 2018 | The analysis on the single items, the review of available data and the Joint Price Monitoring data | Using vulnerability assessment data on the monthly expenditures of an average household of 6 individuals | The Survival Minimum Expenditure Basket (SMEB) for covering 2100 Kcal | 227.7 | - | 227.7 | - | - | - | - | - | | | - | 61 | - | 33.3 | 33.3 | Medium |
| | | | | | | | | | Jordan | | | | | | | | | | | |
| 20 | UNHCR [f], 2019 | Based on the nutritional value that key commodities provide | Data from the parents of children attending formal schools, extreme/overrated values | SMEB for a daily diet of 2100 Kcal (11.6 g of protein and 19.2 g of fat per person/per day) | 200 | 50 | 150 | - | 20 | - | 8 | - | 30 | - | 19 | 40 | - | 33 | 33 | Medium |

[a] SCI: Statistical Centre of Iran, [b] FGDs: Focus Group Discussions, [c] HH: Household, [d] MEB: Minimum Expenditure Basket, [e] SMEB: Survival Minimum Expenditure Basket, [f] UNHCR: UN Refugee Agency, [g] LACON diet: Locally Appropriate Cost-Optimized Nutritious diet, [h] CMWG: Cash and Markets Working Group, [i] CoRD: The Cost of a Recommended Diet, [j] FBDGs: Food-based dietary guidelines.

**Table 2.** Characteristics of healthy, sustainable or survival minimum expenditure food baskets recommended by international organizations.

| No. | Author, Date | Designing Method | Source of Data/ Population | Results Based on/Energy Provided | Components of Food Basket (Food Groups in g) | | | | | | | | | | | | | | | Quality of the Study |
|---|---|---|---|---|---|---|---|---|---|---|---|---|---|---|---|---|---|---|---|---|
| | | | | | Bread | Macaroni/ Pasta | Rice | Potato (Starchy Veg) | Vegetable | Fruit | Milk &Dairy Products | Red Meat | Poultry | Fish | Egg | Legume (Pulse) | Nut | Fat&Oil | Sugar &Sweet | |
| **Nutrition Guidelines** | | | | | | | | | | | | | | | | | | | | |
| 1 | UN Refugee Agency (UN-HCR), 1995 | Recommended ration for the classic full food basket | Refugees or displaced people | The Survival Minimum Expenditure Basket to cover 2261 Kcal/ day | 400 (Cereal: maize) + 100 (Fortified Cereal Blend: corn soya blend) | - | - | - | - | - | - | - | - | - | - | 60 | - | 25 | 15 | Medium |
| **Minimum expenditure basket** | | | | | | | | | | | | | | | | | | | | |
| 2 | World Food Program (WFP), 2020 | Hybrid approach (mix of an expenditure-based and a rights-based approach) | Crisis-affected populations | Scaling to 2100 kcal per person per day, with 10–12 percent of daily energy intake from protein and 17 percent from fats | 424 | | - | | 182 | 6 | 1 | 81 | | | - | 24 | - | 33 | 6 | High |
| **Global Healthy Reference Diet** | | | | | | | | | | | | | | | | | | | | |
| 3 | Willet et al., 2019 | Meeting all requirements of 20 essential nutrients | Generally healthy individuals aged 2 years and older | A healthy 60 kg woman at 30 years old, in energy balance at 2503 kcal per day | 232 | | | 50 (0–100) | 300 (200–600) | 200 (100–300) | 250 (0–500) | 14 (0–28) | 29 (0–58) | 28 (0–100) | 13 (0–25) | 75 (0–100) | 50 (0–75) | 51.8 (20–91.8) | 31 (0–31) | High |

### 3.3. Historical Perspective and Constructing Methods

Based on the identified documents, only three countries (Iran, Lebanon, and Yemen) have had edits on one food basket based on price changes and updated dietary guidelines. In Iran, the most widely used National FB was presented in 2013 [42] based on the amendments of two previous ones [38,39], and the final stages of revising this FB are underway. In Lebanon, estimation of the needs through expenditure baskets for Syrian Refugees was first introduced in 2014 [48] andwas edited twice, in 2016 [46] and 2020 [47]. In addition, 2006 Pakistan's FB and 2012 Yemen's Food Security and Agriculture Cluster (FSAC) minimum FB were mentioned in the FB documents, which could not be accessed in the searches.

Linear & goal programming was used by 9 studies to estimate the FB groups [39–42,44,49,50,60,61]; however, focus group discussion ($n$ = 5) [46–48,51,53], mean requirement of energy, protein, and key nutrients ($n$ = 5) [38,42,54–56], and market assessment ($n$ = 1) [52] also contribute to FB studies in the region. Most of the baskets are uprooted from the national income and expenditure surveys [37,40,41,46–49,55], market surveys (to identify the healthy and lowest cost diet that meets the needs for energy and micronutrients) [50,51], food consumption data [38,39], food balance sheets [42], the nutritive value (provided by pivotal food goods) [56]. In addition to suggesting a healthy or low-cost FB for the population, some studies have presented their baskets based on age–sex groups [41,42], income deciles [40], or a vulnerable household [51,52].Four countries (Lebanon [46–48], Syria [54], Iraq [55], and Jordan [56]) out of the seven countries developing FBs in the region have used the typical starting point for establishing a minimum expenditure basket (MEB) to estimate the cost of acquiring enough food to meet energy requirements, usually 2100 Kcal per person per day based on the Sphere Standard [62], the World Bank's Handbook for Poverty and Inequality, the WFP (World Food Program) guidance note for minimum expenditure baskets [58,63], or The United Nations High Commissioner for Refugees (UNHCR) [57]. Others often have used energy estimates based on their country's age–gender composition, which generally leads to higher estimates in the range of 2300 to 2800 Kcal. Exception for a few FBs, such as Yemen's proposed SMB [52,53], which provides 80% of energy and micronutrients, other baskets claim to provide most of the required micronutrients.

### 3.4. Data Synthesis

Although it is assumed that the EMR countries have similar eating habits and patterns, suggested amounts of food items in the FBs are very variable. Survival and minimum baskets have lower quantities of food groups compared to optimum and sustainable baskets. The lowest energy and number of food groups in the FB were found in Yemen.

Bread, Macaroni/pasta, Rice, and Starchy vegetables: The bread and cereal group are presented together in the Global Healthy Reference Diet and several other baskets. However, the recommended range is very different. Bread as the main staple food of the region has been included in all baskets (130–357g per capita per day in moderate to high-quality studies). The coming food groups are rice (about 100 g), Macaroni/pasta, and Starchy vegetables (potato), respectively.

Red meat, Poultry, Fish, and Egg: With a few exceptions, the consumption of sea foods in this region is not common; as a result, in some baskets, chicken, fish, and in some cases, meats are presented as a whole.

Legumes and Nuts: In most baskets of the region, legumes are estimated in combination with nuts. The lowest and the highest recommended values belong to Iran (18 g) and Pakistan (137.6 g), respectively; whereas FBs established by the international organizations (Table 2) recommend 24–75 g legumes. Nuts were mentioned as a separate food group only in global recommendations (0–75 g) and a few of Iran's baskets (7.4–20 g).

Fat and oil, Sugar and sweets: The distance between the minimum and maximum limits is estimated to be very large; however, by removing the outlier numbers, the range is almost close to the global recommendations.



*3.5. Quality of the Reviewed Studies*

Most of the studies included in the review were of the medium to low quality mostly due toafailure to identify confounders and use strategies to deal with them. Figures such as 11 g of sugar in Iran [41,45] and 3 g of fat and oil in Pakistan [49], which are far from other estimates and the usual consumption of society, indicate the weakness of the estimates, too.

## 4. Discussion

The present systematic review found 20 estimations of FBs in EMR, of which about half of them provide the optimum FBs [37–42,44,45,49,51,61] and the other half provide the Survival Minimum Expenditure Baskets (SMEB) [38,46–48,54–56]. A few studies have also considered principles of sustainability in designing the baskets [37,44]. According to the recommendations based on sustainability in recent years, less consumption of red meat had been proposed due to its environmental effects. The poultry group, along with eggs, plays an important role in supplying animal protein in the region. According to the Mediterranean diet, consumingtwoservingsof fish and other seafoodsweekly along with re-ducing the consumption of red meat and saturated fat are of importance intheprevention of non-communicable diseasesand can be a practical and effective choice among the available practical dietary strategies to achieve the maximal benefits for human and environmental health [58–61]. Therefore, it is better to put them as a separate group in recommended FBs for more emphasis.

Despite the traditional production of some nuts (e.g., walnuts, almonds, and pistachios) in the Middle Eastern countries, their recommendation to the public based on regional dietary guidelines [64] is not possible due to their high price. However, considering the dietary habits of this region and the variety of traditional foods that are cooked with legumes, more than 30 g daily is suggested. Since most of the oils and fats consumed are not of high quality in terms of fatty acid content, and high trans-fatty acids in food products are still a nutritional problem in these countries [65], consumption of more than 30–40 g of oil and fat will not be appropriate for the region.

Countries such as Lebanon [47] have reduced their sugar intake recommendations in recent years in line with WHO guidelines, ref. [66] implying the limitation offree sugars intake to less than 10% of total energy intake, while the amount of sugar in Iran's recent FBs has been estimated at a higher level by considering both conditions of cost minimization and stability maximization [44,67]. The biggest contributors to sugar consumptionin children and adolescents of this region have beensugar-sweetened beverages, biscuits, and chocolates [68]. It seems that in adults, consuming sugar and sweets along with hot drinks contributes the most to free sugar consumption.

IntheWestern Mediterranean region, updating the Spanish Healthy Food Reference Budget (SHFRB) to include the dimensionof sustainability resulted in a sustainable basket cheaper than current recommendations [69]. A shift towards plant-based foods, especially whole grains, legumes, and nuts, along withareduction in the level of meat with the exception of fish, is a characteristic of these baskets, which is consistent with the EAT-Lancet report [70,71]. Some of these sustainable baskets consider proximity, packaging, and seasonalitycriteria to stress environmentally friendly food consumption, too [69].

Estimation of FBs provides amounts from the categories of nutrient-dense foods and beverages in purchasable forms, as well as associated costs within calorie limits to support a healthy and nutritious diet, which can help individuals achieve and maintain good health and reduce the risk of chronic diseases throughout all stages of life. The process of developing the FBs in the developed countries can be described in two phases, each with multiple steps: (1) Identifying and preparing data sources, developing the Modeling Categories, and establishing the inputs and constraints, and (2) Running the optimization model and constructing the minimum FB [13]. A minimum expenditure basket is constructed by estimating the cost of acquiring adequate food and adding the cost of other essential non-food expenditures [58].

It is, generally, attemptedto design FBs in accordance withthe common consumption and current price of food items to ensure and increase their public acceptability. However, qualitative and quantitative studies on the acceptability of the suggested FBs in different income groups in different countries could be a topic for future investigations. Further research investigating other barriers towards compliance with Food-based dietary guidelines (FBDGs) among consumers would allow more targeted implementation and promotion of guidelines [72].

Exclusion from the review was mostly due to the lack of information about FBs or lack of access to it. Including some food items in the basket (e.g., processed meat and hydrogenated oils) based on the dietary patterns of the studied communities lowered the quality of one of the reviewed studies [37], whereas neglecting the current diets led to a decrease in the quality of others despite using high statistical analysis methods [44]. The cooperation of experts in various fields like food economics, health education, and nutrition can help reduce the problems of baskets and make them more comprehensive.

Strengths and limitations: Although this study has aimed to gather and summarize studies conducted on food baskets in EMR to provide evidence in a common FB in this region, this review did not result in such evidence due to the heterogeneity of the available data. However, it provides a comprehensive assessment of the food baskets of the region by country.

## 5. Conclusions

Estimation methods of 20 healthy or minimum FB studies found in EMR were different in all criteria and most of them are not fully consistent with the recommendations of the current guidelines. Study results are not comparable due to differences in the tools, protocols, and methods; hence, there is a need for a standardized regional approach. Assessment of the price and affordability of healthy (recommended) and current diets would provide more robust and meaningful data to reform health and fiscal policies in EMR.

Achieving overall health goals and societal outcomes of recommended FBs will depend on the efforts of nutritional health boards in collaboration with many other community partners, including non-governmental organizations, local and municipal governments, government-funded agencies, and the private sector. The health of individuals and communities in EMR is significantly affected by complex interactions between socio-economic factors, the physical environment, and individual behaviors and conditions.

**Author Contributions:** F.M.-N. and M.P. contributed to all phases of the review; A.H., M.A. and A.A.-J. mostly contributed to conceptualizing and reviewing the documents; and M.-R.K., F.E. and G.F. contributed to data gathering and extracting. All authors have read and agreed to the published version of the manuscript.

**Funding:** This work was supported by the National Nutrition and Food Technology Research Institute (NNFTRI, 00.28238), Faculty of Nutrition Sciences and Food Technology; Shahid Beheshti University of Medical Sciences, Tehran, Iran.

**Institutional Review Board Statement:** This study was conducted according to the PRISMA systematic review guidelines and approved by the Ethics Committee of National Nutrition and Food Technology Research Institute (IR.SBMU.NNFTRI.REC.1400.050).

**Informed Consent Statement:** Not applicable.

**Data Availability Statement:** Not applicable.

**Acknowledgments:** The authors would like to thank the NNFTRI for the funding support, and all the researchers, who made this review possible with their valuable research on food baskets in the Eastern Mediterranean Region. We also thank Mansoor Ranjbar and Marzieh Soleymaninejad from the regional office of the WHO in Iran, and Ibrahim Parvin, for their detailed review of the manuscript.

**Conflicts of Interest:** The authors declare no conflict of interest.The funders had no role in the design of the study; in the collection, analyses, or interpretation of data; in the writing of the manuscript; or in the decision to publish the results.

## Appendix A

**Table A1.** Search strategy of food baskets recommended in the Eastern Mediterranean Region.

| PubMed |
| --- |
| ((healthy OR standard OR affordable OR minimized OR adequate OR "low cost" OR optimized OR sustainable OR optimum OR reference OR minimum OR survival OR nutritious OR thrifty OR basic OR balanced) AND ("Food Basket" [Title/Abstract] OR "food plan" [Title/Abstract] OR diet [Title/Abstract] OR "diet plan" [Title/Abstract] OR "dietary advice" [Title/Abstract] OR "food plan" [Title/Abstract] OR "food aid" [Title/Abstract])AND ("Afghanistan " [Title/Abstract] OR "Bahrain" [Title/Abstract] OR "Djibouti" [Title/Abstract] OR "Egypt" [Title/Abstract] OR "Iran" [Title/Abstract] OR "Iraq" [Title/Abstract] OR "Jordan " [Title/Abstract] OR "Kuwait" [Title/Abstract] OR "Lebanon" [Title/Abstract] OR "Libya" [Title/Abstract] OR "Morocco" [Title/Abstract] OR "Palestine" [Title/Abstract] OR "Oman" [Title/Abstract] OR "Pakistan " [Title/Abstract] OR "Qatar" [Title/Abstract] OR "Saudi Arabia " [Title/Abstract] OR "Somalia" [Title/Abstract] OR "Sudan" [Title/Abstract] OR " Syria" [Title/Abstract] OR "Tunisia" [Title/Abstract] OR "United Arab Emirates" [Title/Abstract] OR "Yemen " [Title/Abstract] OR "Middle East " [Title/Abstract])) |
| Scopus |
| TITLE-ABS-KEY ((healthy OR standard OR affordable OR minimized OR adequate OR "low cost" OR optimized OR sustainable OR optimum OR reference OR minimum OR survival OR nutritious OR thrifty OR basic OR balanced) AND ("food basket" OR diet OR "diet plan" OR"dietary pattern" OR"dietaryadvice"OR"food plan" OR "food aid")AND(afghanistanORbahrain OR djiboutiORegyptORiranORiraqORjordanORkuwaitOR-lebanonORlibyaORmoroccoORpalestineORomanORpakistanORqatarORsaudiAND arabiaORsomaliaORsudanORsyriaORtunisiaOR"united arabemirates"ORyemen OR "Middle East")) |
| ISI/WOS |
| ((healthy OR standard OR affordable OR minimized OR adequate OR "low cost" OR optimized OR sustainable OR optimum OR reference OR minimum OR survival OR nutritious OR thrifty OR basic OR balanced) AND ("food basket" OR diet OR "diet plan" OR "dietary pattern" OR "dietary advice"OR"food plan" OR "food aid") AND (afghanistanORbahrain OR djiboutiORegyptORiranORiraqORjordanORkuwaitORlebanonOR-libyaORmoroccoORpalestineORomanORpakistanORqatarORsaudiAND arabiaORsomaliaORsudanORsyriaORtunisiaOR"unitedarabemirates"ORyemen OR "Middle East"))Timespan= All years. ANDIndexes: SCI-EXPANDED, SSCI, A&HCI, CPCI-S, CPCI-SSH, BKCI-S, BKCI-SSH, ESCI, CCR-EXPANDED, IC= All years. |

## Appendix B

**Table A2.** PRISMA 2020 Checklist.

| Section and Topic | Item # | Checklist Item | Location Where Item Is Reported |
| --- | --- | --- | --- |
| Title | 1 | Identify the report as a systematic review. | Title Page, lines 1–2 |
| ABSTRACT | | | |
| Abstract | 2 | See the PRISMA 2020 for Abstracts checklist. | Page 1, lines1–23 |
| INTRODUCTION | | | |
| Rationale | 3 | Describe the rationale for the review in the context of existing knowledge. | Page 2, lines 27–45 |
| Objectives | 4 | Provide an explicit statement of the objective(s) or question(s) the review addresses. | Page 2, 3, lines 46–62 |

**Table A2.** *Cont.*

| Section and Topic | Item # | Checklist Item | Location Where Item Is Reported |
|---|---|---|---|
| METHODS | | | |
| Eligibility criteria | 5 | Specify the inclusion and exclusion criteria for the review and how studies were grouped for the syntheses. | Pages 4, 5, lines 77–89, 99 |
| Information sources | 6 | Specify all databases, registers, websites, organisations, reference lists and other sources searched or consulted to identify studies. Specify the date when each source was last searched or consulted. | Page 3, lines 64–69 |
| Search strategy | 7 | Present the full search strategies for all databases, registers and websites, including any filters and limits used. | Appendix A, search strategies |
| Selection process | 8 | Specify the methods used to decide whether a study met the inclusion criteria of the review, including how many reviewers screened each record and each report retrieved, whether they worked independently, and if applicable, details of automation tools used in the process. | Page 4, lines 77–89 |
| Data collection process | 9 | Specify the methods used to collect data from reports, including how many reviewers collected data from each report, whether they worked independently, any processes for obtaining or confirming data from study investigators, and if applicable, details of automation tools used in the process. | Page 4, lines 77–91 |
| Data items | 10a | List and define all outcomes for which data were sought. Specify whether all results that were compatible with each outcome domain in each study were sought (e.g., for all measures, time points, analyses), and if not, the methods used to decide which results to collect. | Page 5, lines 99–103 |
| | 10b | List and define all other variables for which data were sought (e.g., participant and intervention characteristics, funding sources). Describe any assumptions made about any missing or unclear information. | - |
| Study risk of bias assessment | 11 | Specify the methods used to assess risk of bias in the included studies, including details of the tool(s) used, how many reviewers assessed each study and whether they worked independently, and if applicable, details of automation tools used in the process. | Pages 4, 5, lines 82–83; 90–98 |
| Effect measures | 12 | Specify for each outcome the effect measure(s) (e.g., risk ratio, mean difference) used in the synthesis or presentation of results. | |
| Synthesis methods | 13a | Describe the processes used to decide which studies were eligible for each synthesis (e.g., tabulating the study intervention characteristics and comparing against the planned groups for each synthesis (item #5). | Pages 4, 5, lines 77–89, 99 |
| | 13b | Describe any methods required to prepare the data for presentation or synthesis, such as handling of missing summary statistics, or data conversions. | - |
| | 13c | Describe any methods used to tabulate or visually display results of individual studies and syntheses. | - |

**Table A2.** *Cont.*

| Section and Topic | Item # | Checklist Item | Location Where Item Is Reported |
|---|---|---|---|
| | 13d | Describe any methods used to synthesize results and provide a rationale for the choice(s). If meta-analysis was performed, describe the model(s), method(s) to identify the presence and extent of statistical heterogeneity, and software package(s) used. | - |
| | 13e | Describe any methods used to explore possible causes of heterogeneity among study results (e.g., subgroup analysis, meta-regression). | - |
| | 13f | Describe any sensitivity analyses conducted to assess robustness of the synthesized results. | - |
| Reporting bias assessment | 14 | Describe any methods used to assess risk of bias due to missing results in a synthesis (arising from reporting biases). | - |
| Certainty assessment | 15 | Describe any methods used to assess certainty (or confidence) in the body of evidence for an outcome. | - |
| RESULTS | | | |
| Study selection | 16a | Describe the results of the search and selection process, from the number of records identified in the search to the number of studies included in the review, ideally using a flow diagram. | Pages 5, lines 105–116 |
| | 16b | Cite studies that might appear to meet the inclusion criteria, but which were excluded, and explain why they were excluded. | Pages 5, 6, lines 117–125 |
| Study characteristics | 17 | Cite each included study and present its characteristics. | Tables 1 and 2 |
| Risk of bias in studies | 18 | Present assessments of risk of bias for each included study. | The last column of Tables 1 and 2 |
| Results of individual studies | 19 | For all outcomes, present, for each study: (a) summary statistics for each group (where appropriate) and (b) an effect estimate and its precision (e.g., confidence/credible interval), ideally using structured tables or plots. | Table 1; Pages 6,7, lines 126–176 |
| Results of syntheses | 20a | For each synthesis, briefly summarise the characteristics and risk of bias among contributing studies. | Page 10, lines 177–181 |
| | 20b | Present results of all statistical syntheses conducted. If meta-analysis was done, present for each the summary estimate and its precision (e.g., confidence/credible interval) and measures of statistical heterogeneity. If comparing groups, describe the direction the effect. | - |
| | 20c | Present results of all investigations of possible causes of heterogeneity among study results. | - |
| | 20d | Present results of all sensitivity analyses conducted to assess the robustness of the synthesized results. | - |
| Reporting biases | 21 | Present assessments of risk of bias due to missing results (arising from reporting biases) for each synthesis assessed. | - |
| Certainty of evidence | 22 | Present assessments of certainty (or confidence) in the body of evidence for each outcome assessed. | - |
| DISCUSSION | | | |
| Discussion | 23a | Provide a general interpretation of the results in the context of other evidence. | Pages 9, lines 193–229 |

**Table A2.** *Cont.*

| Section and Topic | Item # | Checklist Item | Location Where Item Is Reported |
|---|---|---|---|
| | 23b | Discuss any limitations of the evidence included in the review. | Page 10, lines 230–231 |
| | 23c | Discuss any limitations of the review processes used. | Page 10, lines 231–234 |
| | 23d | Discuss implications of the results for practice, policy, and future research. | Page 10, lines 234–239 |
| OTHER INFORMATION | | | |
| Registration and protocol | 24a | Provide registration information for the review, including register name and registration number, or state that the review was not registered. | The review was not registered |
| | 24b | Indicate where the review protocol can be accessed, or state that a protocol was not prepared. | Page 11, lines 256–259 |
| | 24c | Describe and explain any amendments to information provided at registration or in the protocol. | - |
| Support | 25 | Describe sources of financial or non-financial support for the review, and the role of the funders or sponsors in the review. | Page 12, lines 263–265 |
| Competing interests | 26 | Declare any competing interests of review authors. | Page 11, line 262 |
| Availability of data, code and other materials | 27 | Report which of the following are publicly available and where they can be found: template data collection forms; data extracted from included studies; data used for all analyses; analytic code; any other materials used in the review. | Page 11, lines 260–261 |

Ref: [31].

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
