# Peer review of "A Systematic Review on Food Baskets Recommended in the Eastern Mediterranean Region"

_sustainability, doi:10.3390/su152014781_

Round 1

Reviewer 1 Report

Title: A Systematic Review on Food Baskets Recommended in the Eastern Mediterranean Region

In the presented manuscript t
he authors aim to identify and characterize food baskets in the Eastern Mediterranean region countries in an attempt to provide a regional set of data and international comparisons.

With some limitations, the study is very good designed. The work is interested and can be accepted for publication in the Sustainability but after minor revision addressing the following points.  

Row 275: “Trans fatty acids” = trans-fatty acids

Discussion section: As comparison of available data is main goal of this review paper, please add some discussion and comparison with food baskets in the Western Mediterranean region.

Conclusions: Why do authors use citations in this section? Conclusion is your one, not cited from previously published studies.

References: Correct the References style as recommended in the guidelines (there are missing data and technical mistakes, use everywhere Journal title abbreviation or full Journal title (e.g., ref. 6 (journal title), 16 (Journal title, vol, page?), …))

Author Response

In the presented manuscript the authors aim to identify and characterize food baskets in the Eastern Mediterranean region countries in an attempt to provide a regional set of data and international comparisons.

With some limitations, the study is very good designed. The work is interested and can be accepted for publication in the Sustainability but after minor revision addressing the following points.  

Thanks for encouraging and constructive comments in order to improve our manuscript.

Row 275: “Trans fatty acids” = trans-fatty acids

It was corrected based on the reviewer’s comment.

Discussion section: As comparison of available data is main goal of this review paper, please add some discussion and comparison with food baskets in the Western Mediterranean region.

The following paragraph was added to Discussion based on the reviewers’ comments:

“In Western Mediterranean region, updating the Spanish Healthy Food Reference Budget (SHFRB) to include the dimension of sustainability resulted in a sustainable basket cheaper than current recommendations [73]. A shift towards plant-based foods especially whole grains, legumes, and nuts along with reduction in the level of meat with the exception of fish, is a characteristic of these baskets, which is consistent with the EAT-Lancet report [74, 75]. Some of these sustainable baskets consider proximity, packaging and seasonality criteria to stress environmentally friendly food consumption, too [73].”

Conclusions: Why do authors use citations in this section? Conclusion is your one, not cited from previously published studies.

Agreed, the citations in Conclusion were transferred to Discussion.

References: Correct the References style as recommended in the guidelines (there are missing data and technical mistakes, use everywhere Journal title abbreviation or full Journal title (e.g., ref. 6 (journal title), 16 (Journal title, vol, page?), …))

All references were checked and corrected.

Reviewer 2 Report

This systematic review was conducted to identify and characterize food baskets in the Eastern Mediterranean region (EMR) countries.

The manuscript deals with a very important topic and need to be implemented:

I have some comments:

Page 15 lines 262-66:  

it would be important to insert a comment on the impact of Med Diet on health and on the prevention of non-communicable chronic diseases. Intake of red meat and saturated fat is associated with an increased risk and mortality from cardiovascular disease and some forms of cancer. quotes:

-       Damigou E, Kouvari M, Chrysohoou C, Barkas F, Kravvariti E, Pitsavos C, Skoumas J, Michelis E, Liberopoulos E, Tsioufis C, Sfikakis PP, Panagiotakos DB; ATTICA Study Group. Lifestyle Trajectories Are Associated with Incidence of Cardiovascular Disease: Highlights from the ATTICA Epidemiological Cohort Study (2002-2022). Life (Basel). 2023 May 8;13(5):1142. doi: 10.3390/life13051142. PMID: 37240787; PMCID: PMC10222365

-       Truzzi, M.L. Puviani, M.B Mediterranean Diet as a model of sustainable, resilient and healthy diet. Progress in Nutrition, 2020, 22(2) doi: 10.23751/pn.v22i2.8632

-       Giosuè A, Riccardi G, Antonelli M. Maximizing cardiovascular benefits of fish consumption within the One Health approach: Should current recommendations be revised? Nutr Metab Cardiovasc Dis. 2023 Jun;33(6):1129-1133. doi: 10.1016/j.numecd.2023.03.019. Epub 2023 Mar 28. PMID: 37087360.

Page 16 lines 278-82

a comment on sugar free consumption is needed at this discussion point. What is the usual consumption of free sugar? is it added to hot drinks? Why has Lebanon given indications to reduce it?  Free sugar habits vary in different geographic areas so you need to specify. This comment can also be included in introduction

no comment to make

Author Response

This systematic review was conducted to identify and characterize food baskets in the Eastern Mediterranean region (EMR) countries.

The manuscript deals with a very important topic and need to be implemented:

 Thanks for encouraging and constructive comments in order to improve our manuscript.

I have some comments:

Page 15 lines 262-66:  

it would be important to insert a comment on the impact of Med Diet on health and on the prevention of non-communicable chronic diseases. Intake of red meat and saturated fat is associated with an increased risk and mortality from cardiovascular disease and some forms of cancer. quotes:

-       Damigou E, Kouvari M, Chrysohoou C, Barkas F, Kravvariti E, Pitsavos C, Skoumas J, Michelis E, Liberopoulos E, Tsioufis C, Sfikakis PP, Panagiotakos DB; ATTICA Study Group. Lifestyle Trajectories Are Associated with Incidence of Cardiovascular Disease: Highlights from the ATTICA Epidemiological Cohort Study (2002-2022). Life (Basel). 2023 May 8;13(5):1142. doi: 10.3390/life13051142. PMID: 37240787; PMCID: PMC10222365

-       Truzzi, M.L. Puviani, M.B Mediterranean Diet as a model of sustainable, resilient and healthy diet. Progress in Nutrition, 2020, 22(2) doi: 10.23751/pn.v22i2.8632

-       Giosuè A, Riccardi G, Antonelli M. Maximizing cardiovascular benefits of fish consumption within the One Health approach: Should current recommendations be revised? Nutr Metab Cardiovasc Dis. 2023 Jun;33(6):1129-1133. doi: 10.1016/j.numecd.2023.03.019. Epub 2023 Mar 28. PMID: 37087360.

 The following explanation and related references were added to page 15 based on the reviewers comment:

“According to the Mediterranean diet, consuming two servings of fish and other see foods weekly along with reducing the consumption of red meat and saturated fat are of importance in prevention of non-communicable diseases and can be a practical and effective choice among the available practical dietary strategies to achieve the maximal benefits for human and environmental health [58-61].”

 Page 16 lines 278-82

a comment on sugar free consumption is needed at this discussion point. What is the usual consumption of free sugar? is it added to hot drinks? Why has Lebanon given indications to reduce it?  Free sugar habits vary in different geographic areas so you need to specify. This comment can also be included in introduction

The following statements were added to Discussion on free sugar intake based on the reviewer comment:

“The biggest contributors to sugar consumption in children and adolescents of this region were sugar-sweetened beverages, biscuits, and chocolates [72]. It seems that in adults, consuming sugar and sweets along with hot drinks contributes the most to free sugar consumption.”
